# Effects of Trimetazidine on Right Ventricular Function and Ventricular Remodeling in Patients with Pulmonary Artery Hypertension: A Randomised Controlled Trial

**DOI:** 10.3390/jcm12041571

**Published:** 2023-02-16

**Authors:** Hugo E. Verdejo, Adolfo Rojas, Camila López-Crisosto, Fernando Baraona, Luigi Gabrielli, Vinicius Maracaja-Coutinho, Mario Chiong, Sergio Lavandero, Pablo F. Castro

**Affiliations:** 1Advanced Center for Chronic Diseases (ACCDiS), Division Enfermedades Cardiovasculares, Facultad de Medicina, Pontificia Universidad Católica de Chile, Santiago 8330077, Chile; 2Advanced Center for Chronic Diseases (ACCDiS), Facultad de Ciencias Químicas y Farmacéuticas & Facultad de Medicina, Universidad de Chile, Santiago 8380492, Chile; 3Department of Internal Medicine, Cardiology Division, University of Texas Southwestern Medical Center, Dallas, TX 75390, USA

**Keywords:** pulmonary artery hypertension, right ventricular failure, trimetazidine, fatty acid beta-oxidation inhibitor, 6 min walk test, pro-BNP

## Abstract

Background: Pulmonary artery hypertension (PAH) is a chronic and progressive disease. Although current therapy has improved the disease prognosis, PAH has a poor survival rate. The key feature leading to disease progression and death is right ventricular (RV) failure. Methods and results: We assessed the role of trimetazidine, a fatty acid beta-oxidation (FAO) inhibitor, in right ventricular function, remodeling, and functional class in PAH patients, with a placebo-controlled double-blind, case-crossover trial. Twenty-seven PAH subjects were enrolled, randomized, and assigned to trimetazidine or placebo for three months and then reallocated to the other study arm. The primary endpoint was RV morphology and function change after three months of treatment. Secondary endpoints were the change in exercise capacity assessed by a 6 min walk test after three months of treatment and the change in pro-BNP and Galectin-3 plasma levels after three months. Trimetazidine use was safe and well-tolerated. After three months of treatment, patients in the trimetazidine group showed a small but significant reduction of RV diastolic area, and a substantial increase in the 6 min walk distance (418 vs. 438 mt, *p* = 0.023), without significant changes in biomarkers. Conclusions: A short course of trimetazidine is safe and well-tolerated on PAH patients, and it is associated with significant increases in the 6MWT and minor but significant improvement in RV remodeling. The therapeutic potential of this drug should be evaluated in larger clinical trials.

## 1. Introduction

Pulmonary artery hypertension (PAH) is a rare disease with a reported prevalence of 5–25 cases per 1 million people [1]. PAH is characterized by a progressive increase of pulmonary vascular bed resistance because of structural and functional changes in the pulmonary vessels, including vasoconstriction and vascular remodeling, which leads to right ventricular hypertrophy and heart failure [2]. The key feature leading to disease progression and death is right ventricular (RV) dysfunction secondary to RV ischemia, cardiac metabolism abnormalities, and RV failure to adapt to high pulmonary artery pressures [3,4]. Predictors of a poor prognosis include advanced functional class, poor exercise capacity as measured by the 6 min walk distance (6MWD) or a cardiopulmonary exercise test, high right atrial (RA) pressure, significant right ventricular (RV) dysfunction, evidence of RV failure, low cardiac index, elevated brain natriuretic peptide (pro-BNP), and underlying diagnosis of scleroderma spectrum of diseases [3].

Although the pathophysiology of the disease is not thoroughly understood, recent findings suggest that mitochondrial metabolic changes may provide a common background for disease progression [5]. PAH is characterized by an increased glycolytic rate with impaired glucose oxidation in the pulmonary artery smooth muscle cells (PASMC) in the process of metabolic remodeling, switching to a non-hypoxic, glycolytic phenotype similar to that seen in cancer cells, with increased cell proliferation and resistance to apoptosis [4,5]. Therefore, interventions to restore glucose oxidation can reestablish apoptotic susceptibility, decrease proliferation, and reduce pulmonary vasoconstriction in animal models [6,7].

Based on these findings, we designed a phase II randomized clinical trial to assess the effects and safety of trimetazidine in patients with PAH. We sought to evaluate the effectiveness of fatty acid inhibition in reducing pulmonary vascular resistance and improving right ventricular function, right ventricular remodeling, and functional capacity in PAH patients.

## 2. Methods

### 2.1. Subject Selection

Inclusion criteria: (a) Male and female patients 18 years old or older with symptomatic PAH. (b) PAH belonging to the following subgroups of the updated Dana Point Clinical Classification Group 1. (c) Documented hemodynamic diagnosis of PAH by right heart catheterization, performed any time before the screening. All subjects signed an informed consent form. “The study was conducted according to the guidelines of the Declaration of Helsinki and approved by the Ethics Committee, Hospital Clinico, Faculty of Medicine, P. Catholic University of Chile (ID 1141198, date of approval: 19 March 2014). NIH Clinical Trial Registration Number NCT02102672.

Exclusion criteria: (a) Patients with pulmonary hypertension (PH) in the Updated Dana Point Classification Groups 2–5, (b) moderate or severe obstructive or restrictive lung disease, (c) moderate or severe hepatic impairment (Child-Pugh B and C), (d) documented left ventricular systolic dysfunction, (e) severe renal insufficiency (Serum creatinine > 2.5 mg/dL), (f) patients receiving any investigational drugs within one month before the baseline visit, (g) acute or chronic impairment (other than dyspnea), limiting the ability to comply with study requirements, (h) psychotic, addictive or other disorder limiting the ability to provide informed consent or to comply with study requirements, (i) life expectancy less than 12 months, (j) females who were lactating or pregnant or plan to become pregnant during the study, (k) known hypersensitivity to any of the excipients of the drug formulations.

### 2.2. Randomization

This was a placebo-controlled, double-blind, case-crossover trial. Subjects were randomly assigned to one of two treatment groups after a simple randomization procedure (computerized random number). Group 1 started with trimetazidine 35 mg b.i.d (Servier, Suresnes, France) for three months, followed by a two-week wash-out period, and then was reallocated to a placebo for three months. Group 2 started with a placebo for three months, followed by a two-week wash-out period, and was then reallocated to trimetazidine 35 mg b.i.d for three months.

The trimetazidine and placebo were in pill form and were identical in appearance. The pills were pre-packed in bottles and consecutively numbered for each participant according to their randomization group. An independent pharmacist dispensed either the active drug or placebo according to the randomization group.

Patients, investigators, and clinical staff were kept blind to the participant’s group assignment. Notably, the clinical staff who obtained outcome measurements were not informed of the assigned group of each participant.

### 2.3. Endpoints

The primary endpoint was the change in RV function after three months of treatment. Secondary endpoints were: (a) changes in exercise capacity assessed by 6 min walk test after three months of treatment, (b) changes in pro-BNP and Galectin-1 after three months of treatment.

### 2.4. IBnMyPXEqGFollow-Up

Clinical and laboratory characteristics, including exercise capacity (6 min walk distance and Borg dyspnea index) and serum biomarkers (pro-BNP, Galectin-1), were assessed at baseline, three and six months. Patients were followed by a trained cardiologist monthly to detect any sign of disease worsening. Events of particular interest were: (a) death, (b) hospitalization for worsening of PAH, (c) study drug side effects.

### 2.5. Sample Size

To date, no clinical trial has evaluated trimetazidine’s role in PAH patients. Based on pre-clinical studies as a proof-of-concept [8], we expected an improvement in 10 ± 5% of right ventricular output after three months of treatment compared to placebo. Assuming an alpha error of 5%, two-tailed, and statistical power of 80%, 20 patients needed to be enrolled in a case-crossover design.

### 2.6. Echocardiographic Evaluation

A complete echocardiographic evaluation was performed in all subjects at baseline, 3 and 6 months using a commercially available ultrasound scanner (Vivid 7, GE Healthcare, Milwaukee, WI, USA) with a 2.5-MHz phased array transducer (M4S). Acquired images were analyzed offline with commercially available software (EchoPac BT12, GE Healthcare, Milwaukee, WI, USA). The study included: (1) Right ventricular (RV) geometry and dimensions according to American Society of Echocardiography (ASE) guidelines. Two-dimensional and three-dimensional analyses were used. (2) RV function using two-dimensional and Doppler parameters, including tricuspid annular plane systolic excursion (TAPSE), systolic wave velocity of lateral tricuspid annulus, right time–velocity integral, RV area change, and Tei index (RV performance). (3) Right atrial (RA) dimensions and function using strain and strain rate derived from speckle tracking with an evaluation of inter-atrial asynchrony.

### 2.7. Statistical Analysis

Comparisons between placebo and trimetazidine for three months of treatment were performed using the paired *t*-test or Wilcoxon signed-rank test, as appropriate. For both tests, a 2-sided *p*-value < 0.05 was used as a cutoff to consider statistically significant comparisons. Measures with missing data greater than thirty percent were excluded from the analysis. Meanwhile, the remaining measures with missing data were imputed using k-NN imputation (k = 5) with all non-excluded data. In brief, k-NN imputation takes the k more related patients with known values and finally impute unknown values based on the observed distance [9]. Analyses were performed using the R packages “rstatix” for statistical tests and “VIM” for k-NN imputation [10].

## 3. Results

### 3.1. Study Population

Twenty-seven patients were recruited from the PAH Clinic in our institution, an academic referral center for PAH, from April 2014 to January 2017. Baseline characteristics are described in Table 1. Patients had moderate to severe precapillary hypertension with mPAP 51 ± 9.9 mmHg, pulmonary wedge pressure 10.7 ± 4.3, and pulmonary vascular resistance of 11.9 ± 4.6 WU with preserved cardiac index (2.2 ± 0.5 L/min/m^2^). PAH etiology was idiopathic in most cases.

NYHA classes II and III were equally represented. The mean age was 48 ± 13 years, with a strong female predominance. Subjects’ walking distance in the 6MWT was mildly impaired (402 ± 124 m). Most patients were on sildenafil; one-third of participants were on endothelin receptor antagonist therapy, and 11.1% were on combination therapy, including ERAs and an inhaled prostacyclin analog. The study ended on September 2017 after all patients completed the pre-specified procedures.

### 3.2. Adverse Events

The use of the study medication was associated with mild symptoms, including mild dyspeptic symptoms, bloating (*n* = 5), and headache (*n* = 2). The frequency of these symptoms was similar during the period with the placebo. During follow-up, three patients in the placebo arm and four in the trimetazidine group presented upper respiratory viral infections that only required symptomatic treatment and were unrelated to the study drug or protocol. Clinical impairment, defined as symptomatic worsening requiring treatment adjustment or hospitalization, occurred in five patients using placebo and three using trimetazidine (Table 2). Interestingly, the time to clinical impairment did not differ (42 + 26 vs. 35 + 17, *p* = N.S. for placebo and TMZ, respectively). All patients completed the protocol; none interrupted the study treatment because of adverse events.

### 3.3. Biomarkers

NT-pro BNP, Galectin-3, and hs-Troponin I remained stable during the follow-up. Trimetazidine had no significant effects on blood count, basic biochemical analysis, or lipid profile (Table 2).

### 3.4. Echocardiography

Trimetazidine treatment was associated with a small but significant reduction of RV diastolic area without changes in right ventricular end-diastolic volume and right ventricular end-systolic volume. Right ventricular time-volume integral—a measurement of RV systolic function—as well as mean pulmonary arterial pressure and right atrial diameter remained stable (Table 3).

### 3.5. Exercise Capacity

Trimetazidine treatment significantly improved 6MWD (increase 17 m, *p* = 0.023) (Figure 1), with a slight, non-significant improvement in Borg dyspnea index despite the variability in the patients’ responses. Baseline oxygen saturation remained unchanged (Table 4). No significant changes were observed in NYHA functional capacity.

## 4. Discussion

This randomized controlled trial provides data on the safety and efficacy of trimetazidine 35 mg b.i.d. in patients with PAH. Our results show that trimetazidine reduced right adverse ventricular remodeling and improved 6MWD with a good safety profile.

The potential role of metabolism in the pathogenesis of PAH has been recently acknowledged, proposing that PAH results from a multistep process driven by crucial genes involved in cell metabolism and proliferation [11]. A hallmark of PH models is the development of mitochondrial metabolism abnormalities, mainly the inhibition of pyruvate dehydrogenase, resulting in aerobic glycolysis in both RV myocytes and lung vasculature [12]. This metabolic change has significant physiological effects, decreasing RV contractility and promoting the phenotype change in PASMC [13] and PAEC [14], which become hyperproliferative and apoptosis-resistant [15]. Coincidently, the use of drugs targeting critical metabolic pathways is an emerging area of research in PAH. Restoration of anaerobic glycolysis, either via PDK inhibition [16] or through the activation of the Randle cycle using FAO inhibitors, has shown promising results [17]. For instance, Fang et al. [8] demonstrated that in rats with pulmonary banding-induced PH, FAO is increased in the RV myocytes, and glucose oxidation is conversely decreased. Treatment with FAO inhibitors such as trimetazidine or ranolazine was associated with an increase of 50% in right ventricular output and a marked increase in treadmill distance.

From a clinical standpoint, trimetazidine has shown cardioprotective effects in small series of patients with angina, diabetes mellitus, and heart failure (HF), with a unique safety pattern due to its lack of interference with heart rate, blood pressure, or concurrent therapies. More interestingly, trimetazidine may potentiate the impact of standard treatment: in subjects with ischemic heart disease and left ventricular dysfunction, trimetazidine enhances the beneficial effects of exercise in functional capacity and LVEF [18]. However, large RCTs are still lacking in establishing its role in HF [19]. Concerning its clinical role in PAH, the only evidence available comes from the work of Bayram et al. [20], which showed that the addition of trimetazidine to standard therapy in patients with cor pulmonale (a condition characterized by pulmonary hypertension secondary to chronic lung disease) reduced oxidative stress and BNP. To our knowledge, our work is the first to assess the therapeutic potential of trimetazidine in PAH.

Several other mechanisms may be involved in the beneficial effect of TMZ in PAH beyond the restoration of anaerobic glycolysis via the Randle cycle. TMZ has a known indirect antioxidant effect; a short course of TMZ increases antioxidant activity and decreases oxidative stress [20,21]. Increased lung and right ventricle ROS is a common finding in several PAH models, including hypoxia, monocrotaline toxicity, caveolin-1 knock-out mice, and transgenic Ren2 rats [22]. Dysregulated ROS signaling contributes to endothelial dysfunction, inflammation and likely participates in the mitochondrial metabolic reprogramming observed in PAH, promoting a hyper-proliferative, anti-apoptotic PASMCs phenotype [23].

Trimetazidine may improve mitochondrial function by restoring mitochondrial dynamics. Our group has shown that low-dose trimetazidine increases mitochondrial function by increasing the rate of fused mitochondria in neonatal cardiomyocytes [24]. More recently, we demonstrated that incubating human PASMC with trimetazidine before hypoxia restored mitochondrial potential and respiratory rates and precluded hypoxia-induced PASMC proliferation by a Drp-1-dependent mechanism [25].

A recently published paper by Bobescu et al. [26] showed that adding trimetazidine to optimal medical therapy in acute coronary syndrome was associated with reduced oxidative stress, endothelial dysfunction, inflammation, and acute cardiovascular events. While the underlying mechanism is not well known, it may depend on the ability of trimetazidine to decrease circulating homocysteine levels [27], increase VEGF levels [28], blunt endothelin-1 release [29], or increase endothelial progenitor cells [30].

In several animal models, the overexpression of miR-21 seems critical for right ventricular remodeling accompanying PAH [31]. In isolated RV myocytes, trimetazidine treatment decreased hypoxia-induced apoptosis and increased miR-21 levels; conversely, transfection with a miR-21 inhibitor decreased the protective effect of trimetazidine [32].

### Strengths and Limitations

Despite its small patient sample size, this proof-of-concept pilot study is the largest controlled trial of trimetazidine in PAH patients to date. It will provide valuable data when designing more extensive randomized controlled trials. This study tested a fixed dose of trimetazidine (35 mg b.i.d). While a positive dose-effect relationship in PAH may be possible, it has not been observed in other pathologies; furthermore, larger doses may increase the risk of uncommon adverse effects such as parkinsonism [33]. Due to the short follow-up, relevant outcomes such as time to clinical worsening or mortality were not observed. However, the 6MWD is a surrogate endpoint of prognostic significance; our findings warrant larger studies to assess the therapeutic potential of trimetazidine in PAH patients.

## 5. Conclusions

This study suggests that adding trimetazidine to standard therapy in PAH is well tolerated and may result in a significant increase in exercise capacity, a known prognostic factor in PAH, with an excellent safety profile. Further studies are required to elucidate the role of trimetazidine in PAH therapy.

## Figures and Tables

**Figure 1 jcm-12-01571-f001:**
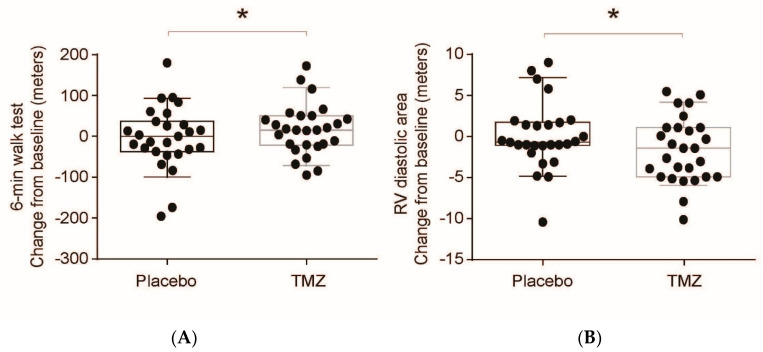
Effect of trimetazidine on exercise capacity and right ventricular (RV) morphology. Twenty-seven PAH subjects were randomized and assigned to trimetazidine (TMZ) or placebo for three months and then reallocated to the other study arm. (**A**) Exercise capacity was determined using 6 min walk distance (6MWD) test. (**B**) RV morphology was assessed by echocardiography. Data were analyzed using paired *t*-test. * *p* < 0.05.

**Table 1 jcm-12-01571-t001:** Baseline demographics and clinical characteristics.

Total, *n*	27
Age, mean ± SD, years	48 ± 13
Female sex, *n*	23
BMI	27.1 (19.1–39.7)
Heart rate	73.4 ± 12.9
Systolic blood pressure	112.2 ± 15.1
Diastolic blood pressure	74.1 ± 10.2
Etiology of PAH, *n*	
Idiopathic	20
Connective tissue disease	4
HIV associated	2
Congenital heart disease associated	1
Time to diagnosis, years	2–30
WHO functional class	
I	1
II–III	12
III	14
Six min walk distance, meters (mean ± SD)	402 ± 124
Hemodynamics,	
Mean PAP (mmHg)	51 (26–80)
CI (L/min/m^2^)	2.2 (1.4–3,3)
PVR (Wood units)	15.2 (3.8–23)
PCWP (mmHg)	11 (4–15)
Medications	
PDE inhibitors (Sildenafil), *n*	23
ERA, *n*	8
Inhaled prostacyclin, *n*	3
Warfarin, *n*	11
Diuretics, *n*	10

BMI: Body mass index, PAP: Pulmonary artery pressure, CI: Cardiac index, PVR: Pulmonary vascular resistance, PCWP: Pulmonary capillary wedge pressure, PDE Phosphodiesterase, ERA Endothelin receptor agonists.

**Table 2 jcm-12-01571-t002:** Reported adverse events.

	Placebo (*n*)	Trimetazidine (*n*)	*p*-Value
Mild adverse events			
Gastrointestinal symptoms	2	5	NS
Headache	1	2	NS
Respiratory tract infections	3	4	NS
Serious adverse events			NS
Clinical worsening	5	3	NS

**Table 3 jcm-12-01571-t003:** Biomarkers and biochemical parameters.

Variable	Baseline	Placebo	Trimetazidine	*p*-Value *
Cardiac markers				
BNP, log (pg/mL)	6.1 ± 1.7	−0.1 ± 0.6	−0.2 ± 0.6	NS
Gal-3, pg/mL	18.1 ± 8.6	−1.7 ± 6.1	−2.0 ± 5.3	NS
p-MYPT/MYPT, UI	1.2 ± 0.9	−0.1 ± 1.7	0.2 ± 1.3	NS
p-EMR/EMR, UI	0.48 ± 0.54	0.02 ± 0.62	0.10 ± 0.51	NS
C-reactive protein, log(mg/dL)	0.91 ± 1.43	−0.22 ± 1.2	−0.27 ± 1.33	NS
hs-troponin T, pg/mL	13.6 ± 10.4	−1.7 ± 3.9	−1.9 ± 3.9	NS
Blood count				
Hematocrit, %	43.2 ± 5.0	−1.1 ± 2.8	−0.4 ± 3.3	NS
Hemoglobin, g/dL	14.2 ± 1.9	−0.3 ± 0.9	0.1 ± 1.2	NS
White cells, count × 10^3^	6.1 ± 2.1	−1.2 ± 2.1	−1.1 ± 2.0	NS
PMN, %	64.3 ± 9.9	−4.3 ± 13.4	−5.05 ± 13	NS
Basophils, %	0.5 ± 0.6	−0.1 ± 0.5	0.1 ± 0.3	NS
Lymphocytes, %	23.7 ± 11.3	5.1 ± 10.6	4.3 ± 11.7	NS
Monocytes, %	7.2 ± 1.9	−0.1 ± 2.0	0.31 ± 1.9	NS
Eosinophils, %	1.9 ± 1.3	0.1 ± 1.1	0.1 ± 1.2	NS
Platelets, count × 10^3^	232 ± 55	−13 ± 27	−14 ± 34	NS
Lipid profile				
Total cholesterol, mg/dL	187 ± 47.9	−14.4 ± 36.3	−4.37 ± 43.5	NS
LDL, mg/dL	115 ± 36.6	−12.3 ± 32.8	−5.07 ± 37.3	NS
HDL, mg/dL	51.5 ± 18	−2 ± 10.2	−0.037 ± 10.4	NS
Biochemical profile				
Triglycerides, mg/dL	116 ± 47.9	−14 ± 50	−10.6 ± 42.2	NS
Creatinine, mg/dL	0.84 ± 0.23	−0.01 ± 0.10	0.02 ± 0.08	NS
Uric acid, mg/dL	5.4 ± 1.23	0.03 ± 1.2	0.2 ± 0.9	NS
AST, UI	26.5 ± 11.5	1.22 ± 17	−2.04 ± 5.64	NS
Calcium, mg/dL	9.3 ± 0.6	−0.007 ± 0.4	0.067 ± 0.4	NS
ALP, UI	89 ± 325	−4 ± 15	2 ± 19	NS
Total bilirubin, UI	0.7 ± 0.4	0.1 ± 0.3	0.1 ± 0.3	NS
Glucose, mg/dL	82.5 ± 12.8	0.333 ± 12.7	2.93 ± 11.7	NS
Ureic nitrogen, mg/dL	15.8 ± 7.92	1.26 ± 5.32	1.07 ± 3.45	NS
Sodium, meq/L	141 ± 1.6	0.6 ± 2.6	0.14 ± 2.3	NS
Potassium, meq/L	4.2 ± 0.3	−0.1 ± 0.3	0.1 ± 0.4	NS

* Data represent changes from baseline.

**Table 4 jcm-12-01571-t004:** Echocardiography and functional parameters.

Variable	Baseline	Placebo	Trimetazidine	*p*-Value *
PAPs, mmHg	77 ± 22	−7 ± 20	−6 ± 16	NS
PAPm, mmHg	45 ± 11	−0.6 ± 11.0	−1.3 ± 9.9	NS
RV time-velocity integral, cm	12.6 ± 3.1	1.1 ± 3.5	1.2 ± 3.1	NS
TVI TR, cm	135 ± 29	−3 ± 27	2 ± 27	NS
RV diastolic area, cm^2^	29 ± 10	0.1 ± 4.1	−1.7 ± 4.0	0.033
RV systolic area, cm^2^	20.5 ± 8.1	−0.2 ± 4.2	−1.4 ± 4.0	NS
RV diastolic volume, cm^3^	94.8 ± 54.8	−1.5 ± 22.8	−7.5 ± 18	NS
RV systolic volume, cm^3^	56 ± 36	−2.6 ± 20	−4.4 ± 14.9	NS
TAPSE, cm	16.6 ± 4.9	−0.02 ± 3.3	0.89 ± 3.53	NS
RA area, cm^2^	26.7 ± 9.9	−0.9 ± 5.3	−1.1 ± 3.1	NS
Oxygen saturation (SaO_2_), %	96 ± 3	−0.15 ± 2.05	0.15 ± 1.8	NS
6-min walk test, meters	421 ± 107	−2.6 ± 77.4	16.9 ± 62.5	0.023
Borg dyspnea index, int	4.4 ± 2.5	0.1 ± 2.3	−0.3 ± 1.8	NS

* Data represent changes from baseline.

## Data Availability

The data presented in this study are available on request from the corresponding author. The data are not publicly available due to data confidentiality reasons.

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
