# Peer review of "Effects of Trimetazidine on Right Ventricular Function and Ventricular Remodeling in Patients with Pulmonary Artery Hypertension: A Randomised Controlled Trial"

_jcm, 2023, doi:10.3390/jcm12041571_

Round 1

Reviewer 1 Report

I had the opportunity to review the work entitled " Effects of trimetazidine on right ventricular function and ventricular remodeling in patients with pulmonary artery hypertension".Although recent efforts, pulmonary hypertension has still a bad prognosis. The article is very well written, designed and organized. The focus of this manuscript is original. I think the results are limited because of the little sample size. In the results you compare dyspnea borg index and what about the nyha class? Were there some improvement and difference between placebo and trimetazidine group? 

Author Response

Query 1. In the results you compare dyspnea borg index and what about the nyha class? Were there some improvement and difference between placebo and trimetazidine group? 

Our response:  No significant changes were observed in NYHA functional capacity (see point 3.5).

Reviewer 2 Report

I read the paper entitled “Plasma CEffects of trimetazidine on right ventricular function and ventricular remodeling in patients with pulmonary artery hyper- 3 tension ” by Hugo E. Verdejo 

I thought the content was very interesting and the paper was well written, but one point I would like to say is that authors should provide more details on adverse events and serious adverse event, perhaps a table showing the frequency in both the trimetazidine group and the control group.

Author Response

Query 1. I would like to say is that authors should provide more details on adverse events and serious adverse event, perhaps a table showing the frequency in both the trimetazidine group and the control group.

Our response: Clinical impairment, defined as symptomatic worsening requiring treatment adjustment or hospitalization, occurred in five patients using a placebo and three using trimetazidine (Table 2). Interestingly, the time to clinical impairment did not differ (42+26 vs. 35+17, p= N.S. for placebo and TMZ, respectively). All patients completed the protocol; none interrupted the study treatment because of adverse events.